# SEAL-RAG: Loop-Adaptive RAG with On-the-Fly Entity Extraction and Fixed-K Gap Repair

## Abstract

We propose **SEAL-RAG**, a training-free, inference-time controller (no fine-tuning of retriever, reranker, or generator) for retrieval-augmented generation that targets multi-hop precision. SEAL executes a fixed retrieval depth $k$ ($k$ = number of passages retrieved per search/micro-query) in a Search $\rightarrow$ Extract $\rightarrow$ Assess $\rightarrow$ Loop cycle. A scope-aware sufficiency check aggregates coverage, typed bridging, corroboration/contradiction, and answerability signals to decide stop vs. targeted repair. At each loop, SEAL performs on-the-fly, entity-anchored (head, relation, tail) extraction, maintains a live entity ledger, and builds a gap specification (missing entities/relations) that triggers one micro-query per repair under the same top-$k$; new candidates are merged via entity-first ranking (prefers passages anchoring those entities) before a single final generation step. On a 1,000-example HotpotQA validation subset in a shared setup, SEAL improves LLM-judged answer correctness by **+10–22 pp** ($k{=}1$) and **+3–13 pp** ($k{=}3$) vs. **SELF-RAG** across backbones, and increases evidence precision@k (gold-title precision) by **+12–18 pp** at $k{=}3$. These gains are statistically significant (chi-square for correctness; paired two-sided $t$-tests for precision/recall/F1; $p{<}0.05$). By keeping $k$ fixed and bounding repairs by $T$ (maximum repair iterations), SEAL yields a predictable, bounded cost profile while replacing distractors rather than broadening context.

## 1 Introduction

Large language models (LLMs) excel across many tasks, yet their reliance on parametric memory makes them prone to hallucination on knowledge-intensive queries (Ji et al., 2023). Retrieval-Augmented Generation (RAG) grounds answers in external evidence (Lewis et al., 2020), but conventional pipelines remain brittle: when the initial fetch misses a crucial bridge, the model rarely repairs it. Simply raising the number of retrieved passages ($k$) or accumulating more context typically increases distractors and cost without closing the missing link (Karpukhin et al., 2020; Khattab & Zaharia, 2020).

Prior extensions aim to mitigate insufficiency. **SELF-RAG** steers retrieval with reflective critique during decoding. Query-expansion and multi-hop strategies broaden coverage through reformulation and chained retrieval (Mao et al., 2021; Trivedi et al., 2020). These approaches can improve recall, but they often rely on multiple retrieve–critique rounds, broaden context, and lack an explicit mechanism to *target and repair* the specific gap that blocks answering.

We propose **SEAL-RAG**, a *training-free, inference-time controller* that directly targets evidence insufficiency under a predictable budget. SEAL holds a *fixed retrieval depth $k$* ($k$ passages per search/micro-query) and executes a bounded repair loop with maximum $T$ iterations: Search $\rightarrow$ Extract $\rightarrow$ Assess $\rightarrow$ Loop. A scope-aware sufficiency check (coverage, typed bridging, corroboration/contradiction, answerability) decides *stop* vs. *targeted repair*. Each repair issues *exactly one micro-query* under the same top-$k$; new candidates *replace* distractors via entity-first ranking, and the system performs a *single final generation* once sufficiency holds. Figure 1 provides an end-to-end overview of the SEAL-RAG pipeline.

**Contributions.** (i) *Entity-anchored gap repair at fixed $k$*: on-the-fly extraction of (head, relation, tail) facts and a live entity ledger yield an explicit *gap specification* that drives one micro-query per loop. (ii) *Scope-aware sufficiency*: a lightweight gate aggregates coverage/bridging/corroboration-contradiction/answerability signals to decide stop vs. repair. (iii) *Replace, don't expand*: candidates

are merged by entity-first ranking to *replace* distractors rather than broaden context, keeping budget predictable with $k$ fixed and loops bounded by $T$.

On a 1,000-example HotpotQA validation slice in a shared environment, SEAL delivers consistent gains *vs. SELF-RAG* across backbones: *LLM-judged Exact Match* (Judge-EM) improves by **+10–22 pp** at $k{=}1$ and **+3–13 pp** at $k{=}3$, and *evidence precision@k* (gold-title precision) rises by **+12–18 pp** at $k{=}3$. Prompts, configs, scripts, and artifacts are provided in the *supplementary material*.

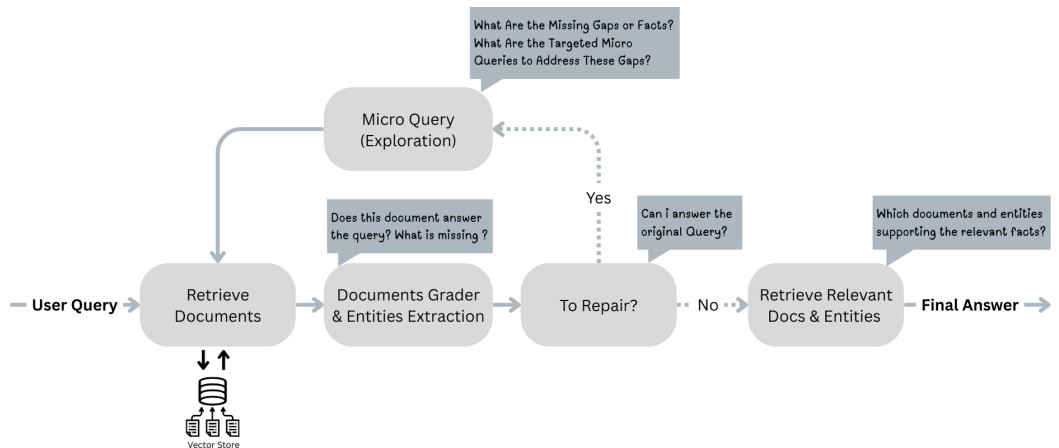

Figure 1: **SEAL-RAG pipeline (Search → Extract → Assess → Loop).** From a user query, initial retrieval (fixed top-$k$) pulls candidates from the vector store. Each loop: **Extract** performs loop-adaptive, entity-first extraction to form a *gap specification*; **Assess** applies scope-aware sufficiency to decide *stop* vs. *repair*. On repair, the *Micro-Query* policy explores targeted queries under the same $k$ (blocklist, stuck detection, pivots). New evidence is integrated via *entity-first ranking*; once sufficient, the system emits the answer under the SEAL rule (2–4 words + citations, or "I don't know").

**Paper organization.** Section 2 reviews related work. Section 3 details SEAL-RAG (loop controller, scope-aware sufficiency, loop-adaptive extraction, entity-first ranking, and the micro-query policy). Section 4 specifies datasets, models, retrieval/indexing, baseline, metrics/judging, and protocol. Section 5 presents main results at $k{=}1$ and $k{=}3$ with per-backbone tables and discussion. Section 6 reports loop-budget ablations and analysis. Section 7 states limitations and threats to validity. Section 8 concludes; anonymized implementation and reproducibility details appear in the *Supplementary Material*.

## 2 RELATED WORK

### 2.1 STANDARD RAG AND LIMITATIONS

Retrieval-Augmented Generation (RAG) grounds LLM outputs in external documents (Lewis et al., 2020), typically via dense retrieval pipelines such as DPR (Karpukhin et al., 2020). However, the standard *retrieve-then-generate* pattern is brittle when the initial top-$k$ misses a bridge fact: increasing $k$ or accumulating more context often inflates distractors and cost without repairing the missing link. SEAL-RAG targets this failure mode by holding $k$ fixed and iteratively *repairing insufficiency* rather than broadly expanding context.

### 2.2 COVERAGE EXPANSION VIA QUERY REFORMULATION AND MULTI-HOP RETRIEVAL

Coverage-oriented methods broaden the candidate pool through reformulation or hierarchical/iterative retrieval, e.g., hypothetical-document embeddings (HYDE) for zero-shot guidance (Gao et al., 2022; Sarthi et al., 2024) and multi-hop strategies that chain retrieval steps. These approaches can lift recall but risk *query drift*, redundancy, and path explosion; they also lack a mechanism to specify *which* factual gap to repair. In contrast, SEAL-RAG derives *gap specifications* from on-the-fly entity

extraction and issues exactly one *micro-query* per loop under the same top-$k$, replacing distractors rather than enlarging the context.

## 2.3 CORRECTIVE AND REFLECTIVE CONTROL IN RAG

**SELF-RAG** integrates retrieval, generation, and critique via reflection tokens, allowing the model to interleave answer drafting with self-assessment and additional retrieval when needed (Asai et al., 2024). This reflective loop improves robustness over single-pass pipelines by encouraging the model to revisit uncertain spans and to re-fetch evidence during decoding.

**Contrast with SEAL-RAG.** While both approaches attempt to correct insufficiency at inference time, their control policies and objectives differ in ways that matter under tight retrieval budgets:

- **Control policy (when/how to retrieve).** SELF-RAG makes retrieval decisions *during generation*, guided by critique tokens that may trigger further fetches. SEAL-RAG instead keeps a *fixed retrieval depth* $k$ and runs an explicit SEARCH→EXTRACT→ASSESS→LOOP controller. Repairs are scheduled *between* generation and only after a sufficiency decision, not interleaved with token emission.

- **What to repair (target).** SELF-RAG's critiques are content- and fluency-aware but do not explicitly localize a *missing entity/relation*. SEAL-RAG performs *entity-anchored extraction* to build a *gap specification* (head/relation/tail, typed bridges), which drives *one* focused micro-query per loop.

- **Stop/continue criterion.** SELF-RAG uses reflection signals as soft guidance during decoding. SEAL-RAG applies a *scope-aware sufficiency* gate that aggregates coverage, typed-bridge, corroboration/contradiction, and answerability signals to decide *stop vs. repair* with thresholds held constant across runs.

- **Evidence management.** SELF-RAG can broaden context as additional passages are pulled during critique. SEAL-RAG *replaces distractors rather than broadening context* via *entity-first ranking*, preserving a fixed-$k$ budget while increasing the fraction of gold titles.

- **Cost profile.** SELF-RAG's critique-triggered retrieval can vary with decoding dynamics. SEAL-RAG bounds inference by design: fixed $k$ and at most $T$ repairs (architecturally $O(k \cdot T)$), yielding predictable cost/latency.

**Other corrective controllers.** CRAG filters/revises candidates but remains within the current pool (Yan et al., 2024); Adaptive-/MAIN-RAG trigger extra retrieval or coordinate roles without localizing a missing fact or enforcing fixed-$k$ replacement (Jeong et al., 2024; Chang et al., 2025). Orthogonal agentic patterns (ReAct, Reflexion) target reasoning/self-improvement rather than gap-targeted repair (Yao et al., 2023; Shinn et al., 2023).

**Empirical context.** Under a shared environment and identical retriever/index/decoding settings, SEAL-RAG yields higher Judge-EM and notably higher Gold-title Precision@k than SELF-RAG at both $k$=1 and $k$=3 (see Sections 5.1 and 5.2). These gains are consistent with SEAL-RAG's design choice to *target and replace* insufficiency at fixed $k$ rather than *broaden* context.

## 2.4 CONTRASTIVE AND RATIONALE-DRIVEN RETRIEVAL

Contrastive re-ranking improves relevance by re-scoring candidates but cannot recover evidence never retrieved, and it adds latency relative to single-stage pipelines. Rationale-driven retrieval uses intermediate reasoning to steer what to fetch next, yet synthesized rationales may drift and broaden context without targeting the specific gap. SEAL-RAG instead performs *scope-aware sufficiency* checks and *entity-anchored* extraction to identify the missing link, then repairs it with one focused micro-query while keeping top-$k$ fixed.

# 3    SEAL-RAG (METHOD)

## 3.1    LOOP CONTROLLER

SEAL-RAG holds a *fixed retrieval depth k* (*k passages per search or micro-query*) and executes a bounded repair loop with maximum $T$ iterations: SEARCH → EXTRACT → ASSESS → LOOP. Given a query $q$, a first-stage retriever $\mathcal{R}$ (top-$k$) (Karpukhin et al., 2020), a reader/generator $\mathcal{G}$, and a sufficiency assessor $\mathcal{S}$, the controller maintains an evolving evidence set $\mathcal{E}_t$, an *entity ledger* $\mathcal{U}_t$, and a blocklist $\mathcal{B}_t$. At each iteration, a sufficiency gate decides *stop* vs. *targeted repair*; generation occurs *once*, after sufficiency holds. Figure 2 shows the execution graph and loopbacks (repair triggers) for the controller nodes.

**Initialization.**    Retrieve $\mathcal{E}_0 \leftarrow \mathcal{R}(q, k)$, extract initial entities $\mathcal{U}_0$ from $\mathcal{E}_0$, and seed $\mathcal{B}_0$ with salient terms already covered to discourage redundancy.

## 3.2    SCOPE-AWARE SUFFICIENCY

The gate aggregates four lightweight signals: (i) **coverage** of question attributes, (ii) **typed bridging** for multi-hop links, (iii) **corroboration/contradiction** across passages, and (iv) **answerability** given the current set. Concretely, $\mathcal{S}$ scores attribute coverage and typed links over $\mathcal{U}_t$, tracks corroboration counts while flagging contradictions, and estimates answerability with calibrated confidence. We declare *sufficient* when all criteria exceed fixed thresholds; otherwise we *repair*. All thresholds, prompts, and calibration details are held constant across backbones and $k \in \{1, 3\}$; full values and templates appear in the *supplementary material*.

## 3.3    LOOP-ADAPTIVE EXTRACTION (ENTITY-ANCHORED)

From retrieved passages, SEAL-RAG performs on-the-fly extraction of (head, relation, tail) facts with supporting spans and updates the *entity ledger*. Rather than summarizing documents, extraction focuses on *windows* around entity mentions and candidate relations implicated by the question. Only *verbatim* triples are stored (Angeli et al., 2015; Bhardwaj et al., 2019) to stabilize grounding and enable direct citation; this ledger supplies candidates to the sufficiency check and subsequent ranking.

## 3.4    GAP SPECIFICATION & ONE MICRO-QUERY

If $\mathcal{S}$ deems $\mathcal{E}_t$ insufficient, extraction compiles a *gap specification*: the *missing attributes/links* and the *entities* they concern, plus blocked/seen terms to avoid redundancy. Each repair loop then issues *exactly one micro-query* under the same top-$k$ targeting the gap; novelty heuristics (blocklists), stuck detection, and safe pivots (Carpineto & Romano, 2012; Lavrenko & Croft, 2001) diversify queries without increasing $k$ (Carbonell & Goldstein, 1998). Operational thresholds and the full micro-query policy are provided in the *supplementary material*.

## 3.5    ENTITY-FIRST RANKING & HANDOFF

Newly retrieved candidates $\Delta\mathcal{E}_{t+1}$ are merged with $\mathcal{E}_t$ via *entity-first ranking*, which prefers passages that anchor ledger entities and resolve missing attributes/links with *verbatim* support. This step *replaces* distractors rather than broadening context; we then re-run sufficiency on the updated set and, once sufficient, execute a single final generation step.

## 3.6    COST PROFILE

Because $k$ is fixed and each repair issues one micro-query (with at most $T$ loops), SEAL-RAG's retrieval/read budget is predictable and bounded (architecturally $O(k \cdot T)$). Per-stage token/latency accounting and any measurements appear in the *supplementary material*.

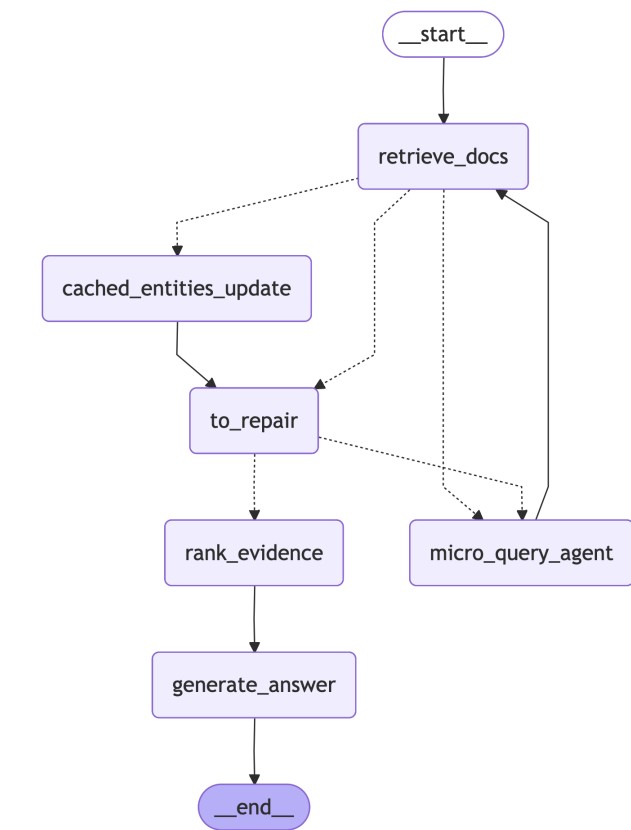

Figure 2: **Execution graph for** SEAL-RAG. Nodes map to stages: `retrieve_docs` (initial retrieval, fixed top-$k$); `cached_entities_update` (entity-anchored extraction); `to_repair` (sufficiency gate); `micro_query_agent` (one micro-query under fixed $k$); `rank_evidence` (entity-first ranking); `generate_answer` (final emission). Solid arrows show the primary path. *Dashed arrows* denote loopbacks triggered by *repair* and indicate steps that may be *executed in parallel* where infrastructure permits (e.g., extraction and ranking over candidate sets), while the controller still enforces fixed-$k$ and a loop budget $T$.

## 4 EXPERIMENTAL SETUP

### 4.1 DATASET

We evaluate on the HotpotQA validation split in the *fullwiki* setting, using a seeded sample of $N{=}1{,}000$ questions for all runs (Yang et al., 2018). HotpotQA mixes *bridge* and *comparison* questions and provides sentence-level supporting facts, enabling evaluation of both answer correctness and evidence quality.

### 4.2 SHARED ENVIRONMENT & MODELS

All methods run in the *same environment*: identical LLM backbones (two capacity tiers), the same dense retriever over Wikipedia, the same fixed retrieval depth $k \in \{1, 3\}$, identical decoding parameters, and the same evaluation judge and metrics. *No model, retriever, or reranker is fine-tuned*; SEAL is a training-free, inference-time controller. Exact model IDs, seeds, package versions, and index metadata are listed in the *supplementary material*.

### 4.3 RETRIEVAL, INDEX, AND ANSWER PROFILE

We use a cosine-similarity vector index with a fixed 1536-d embedding; the retriever returns the top-$k$ passages per query. SEAL's micro-queries *respect this same $k$*. Documents are chunked with a fixed window and normalized uniformly across systems. All methods emit answers under a shared profile: *2–4 words with citations* when supported, otherwise *"I don't know"*.

### 4.4 BASELINE

We compare to **SELF-RAG**, a leading reflective controller for retrieval-augmented generation that reports strong results among *training-free, inference-time* controllers within standard retrieve–read pipelines (Asai et al., 2024). We use it as our primary comparator due to its prominence in this setting. For fairness, both methods use the same backbones, retriever, Wikipedia index, decoding parameters, judge, and metrics; we match the retrieval depth $k$ and bound SELF-RAG's critique-triggered retrieval rounds by the same loop budget as SEAL-RAG's repairs.

### 4.5 METRICS AND JUDGING

Primary metrics (reported in all tables): **Final Answer Correctness (Judge-EM)** from a fixed LLM judge, and **Gold-title Precision/Recall/F1@k** computed over deduplicated retrieved titles versus gold supporting-fact titles. Abstentions (*"I don't know"*) are counted as incorrect for Judge-EM. Metric definitions, title normalization, and the judge rubric are fixed across systems; prompts/configs appear in the *supplementary material*.

**Significance.** We report 95% bootstrap CIs and test SEAL vs. SELF-RAG on the same examples: $\chi^2$ for Judge-EM; paired two-sided $t$-tests for Precision/Recall/F1; Holm–Bonferroni at $\alpha=0.05$. Full statistics, per-seed results, and exact $p$-values are provided in the *Supplementary Material*.

### 4.6 SLICE AND PROTOCOL

Unless stated otherwise, all results share the seeded 1k subset and evaluate at $k \in \{1, 3\}$ while varying the repair-loop budget $L$ (bounded by $T$). This isolates the effect of loop-based repairs under constant $k$ and a shared infrastructure (retriever, index, judge).

## 5 RESULTS

### 5.1 MAIN RESULTS @ $k=1$

*Summary.* Table 1 reports all four metrics at fixed $k=1$ on the seeded 1k HotpotQA slice; methods share identical environments (models, retriever, judge, metrics).

**Key observations @ $k=1$.** Across all backbones, SEAL-RAG improves Judge-EM by **+10–22** pp over **SELF-RAG** and lifts both Precision@1 and Recall@1, consistent with fixed-$k$ loop repairs that *replace* distractors with the needed gold page. F1 gains follow the same pattern.

### 5.2 MAIN RESULTS @ $k=3$

*Summary.* Table 2 reports all four metrics at fixed $k=3$ on the same slice.

**Key observations @ $k=3$.** With three slots, recall rises for both systems, but SEAL-RAG maintains a clear *precision@k* advantage of **+12–18** pp across backbones, yielding **+3–13** pp higher Judge-EM. Where recall dips on the two "mini" backbones, precision gains still translate into equal or higher F1/EM overall.

### 5.3 DISCUSSION

**What fixed-$k$ buys you.** Under small $k$, each retrieved slot is scarce; a distractor directly crowds out a necessary bridge/corroboration page. SEAL's loop keeps $k$ constant and allocates effort to

Table 1: **Main results at** $k=1$ on HotpotQA ($N=1,000$). Metrics are percentages; all settings identical except control logic. $\Delta$ columns show **SEAL−SELF** (same backbone).

| Backbone | Method | Metrics (%) | | | | $\Delta$ (pp) | | | |
|---|---|---|---|---|---|---|---|---|---|
| | | Judge-EM | Prec@k | Rec@k | F1@k | $\Delta$ EM | $\Delta$ Prec | $\Delta$ Rec | $\Delta$ F1 |
| gpt-4o-mini | SELF-RAG | 48 | 61 | 31 | 41 | | | | |
| gpt-4o-mini | **SEAL-RAG** | **62** | **86** | **44** | **58** | **+14** | **+25** | **+13** | **+17** |
| gpt-4o | SELF-RAG | 59 | 75 | 37 | 50 | | | | |
| gpt-4o | **SEAL-RAG** | **73** | **91** | **62** | **72** | **+14** | **+16** | **+25** | **+22** |
| gpt-4.1-mini | SELF-RAG | 49 | 72 | 36 | 48 | | | | |
| gpt-4.1-mini | **SEAL-RAG** | **71** | **87** | **48** | **61** | **+22** | **+15** | **+12** | **+13** |
| gpt-4.1 | SELF-RAG | 63 | 79 | 40 | 53 | | | | |
| gpt-4.1 | **SEAL-RAG** | **73** | **90** | **66** | **74** | **+10** | **+11** | **+26** | **+21** |

Table 2: **Main results at** $k=3$ on HotpotQA ($N=1,000$). $\Delta$ columns show **SEAL−SELF** (same backbone).

| Backbone | Method | Metrics (%) | | | | $\Delta$ (pp) | | | |
|---|---|---|---|---|---|---|---|---|---|
| | | Judge-EM | Prec@k | Rec@k | F1@k | $\Delta$ EM | $\Delta$ Prec | $\Delta$ Rec | $\Delta$ F1 |
| gpt-4o-mini | SELF-RAG | 60 | 66 | 47 | 53 | | | | |
| gpt-4o-mini | **SEAL-RAG** | **69** | **84** | 44 | **57** | **+9** | **+18** | **−3** | **+4** |
| gpt-4o | SELF-RAG | 71 | 76 | 55 | 61 | | | | |
| gpt-4o | **SEAL-RAG** | **77** | **89** | **68** | **75** | **+6** | **+13** | **+13** | **+14** |
| gpt-4.1-mini | SELF-RAG | 64 | 73 | 56 | 61 | | | | |
| gpt-4.1-mini | **SEAL-RAG** | **77** | **86** | 49 | 61 | **+13** | **+13** | **−7** | **0** |
| gpt-4.1 | SELF-RAG | 73 | 79 | 61 | 66 | | | | |
| gpt-4.1 | **SEAL-RAG** | **76** | **91** | **73** | **79** | **+3** | **+12** | **+12** | **+13** |

*targeted replacements*, not breadth-first growth. Empirically, precision@k lifts are substantial at both depths—maximizing at $k=1$ (e.g., +25 pp; Table 1) and remaining consistently high at $k=3$ (+12–18 pp across backbones; Table 2)—with corresponding Judge-EM gains in both settings.

**Effect of backbone strength and $k$.** In our shared setup, SEAL's margin over SELF narrows as base model capacity increases and as $k$ grows. Average Judge-EM gains decrease from $\sim$ +15.0 pp at $k=1$ to $\sim$ +7.8 pp at $k=3$ (per-backbone: 14→9 for 4o-mini; 14→6 for 4o; 22→13 for 4.1-mini; 10→3 for 4.1; Tables 1 and 2). This trend is expected: stronger backbones and larger $k$ reduce initial distractors, leaving less headroom for SEAL's fixed-$k$ replacement to improve precision—yet gains remain consistently positive.

**Why precision matters even when recall dips.** Gold-title@k recall is capped by $k$ when a question needs multiple pages. SEAL's controller explicitly aims to *replace* weaker candidates with gold pages; when it prioritizes the *most critical* missing page (bridge or second entity), Judge-EM can rise even if the auxiliary gold page remains uncovered. This matches HotpotQA's bridge/comparison structure and is visible on the "mini" models at $k=3$.

**Qualitative cases (abridged). Bridge ($k=3$, gpt-4o).** Q: "What is the capital of the U.S. state where the University of Michigan is located?" SELF-RAG retains *Ann Arbor* context and misses the state-capital page; SEAL blocks redundancies, micro-queries "Michigan state capital," swaps in *Lansing* $\Rightarrow$ **Judge-EM=1**. **Comparison ($k=3$, gpt-4.1-mini).** Q: "Who has played for more NBA

teams, Michael Jordan or Kobe Bryant?" SELF-RAG retrieves Jordan twice and lacks Kobe's teams list; SEAL pivots to Kobe's career page and answers correctly.

**Observed failure modes.**   Common errors include (i) bridge entities mentioned only via rare aliases that the index fails to surface, (ii) implicit or temporal attributes (harder for verbatim, entity-anchored extraction), and (iii) borderline paraphrases where the judge may flip the label. These concentrate in long-tail aliases and time-varying facts; representative cases appear in the supplementary.

**Practical guidance.**   Under tight budgets, $k=1$ with a small loop budget ($L \in \{1, 3\}$) captures most of SEAL's gains by swapping a single distractor for the needed bridge page. When latency permits, $k=3$ with the same $L$ yields larger precision margins (more opportunities to replace non-gold titles) and consistently higher Judge-EM. Larger $L$ targets long-tail items with diminishing returns (Section 6.1).

**Statistical confidence.**   All SEAL–SELF differences reported in Tables 1 and 2 are statistically significant under the protocol in Section 4.5 (Holm–Bonferroni, $\alpha=0.05$); detailed $p$-values and 95% CIs appear in the *Supplementary Material*.

**Takeaway.**   Across $k \in \{1, 3\}$ and four backbones, **SEAL-RAG** consistently outperforms **SELF-RAG** on Judge-EM while improving precision under a fixed retrieval budget. The gains align with SEAL's design: entity-anchored extraction, scope-aware sufficiency checks, and micro-query repairs that replace distractors instead of broadening context.

**Reproducibility.**   We release predictions, scripts, and configuration to regenerate all tables/figures from the 1k slice; anonymized artifacts are included in the supplementary for reviewer access.

# 6   ABLATIONS & ANALYSIS

## 6.1   EFFECT OF LOOP BUDGET $L$ AT FIXED $k=1$ (JUDGE-EM)

*Summary.* Table 3 reports *Final Answer Correctness (Judge-EM)* as a function of the repair-loop budget $L$ with retrieval depth fixed at $k=1$.

Table 3: Judge-EM (%) vs. repair-loop budget ($L$) at fixed $k=1$. $\Delta@5$ is the improvement from no repairs ($L=0$) to the maximum budget ($L=5$). Complementary precision/recall curves are provided in the *Supplementary Material*.

|  | **Judge-EM (%) at Loop Budget** $L$ | | | | |
| --- | --- | --- | --- | --- | --- |
| **Backbone** | $L = 0$ | $L = 1$ | $L = 3$ | $L = 5$ | $\Delta$ **vs.** $L = 0$ |
| gpt-4o-mini | 30 | 58 | 61 | 62 | +32 |
| gpt-4.1-mini | 28 | 66 | 70 | 71 | +43 |
| gpt-4o | 32 | 67 | 71 | 73 | +41 |
| gpt-4.1 | 25 | 63 | 69 | 73 | +48 |
| **Average** | **29** | **64** | **68** | **70** | **+41** |

**Motivation.**   This isolates the causal contribution of SEAL-RAG's *repair loop* under a fixed retrieval budget. With $k=1$ held constant, changes across $L$ reflect loop policy (gap detection $\rightarrow$ micro-queries $\rightarrow$ replacements), not larger context windows or more retrieved passages.

**Design & controls.**   We vary only $L \in \{0, 1, 3, 5\}$ and keep the full environment identical to Section 4.2: same backbones, index snapshot, seeded 1k slice, judge, and metrics. $L=0$ is single-pass (no repairs). For $L>0$, each repair issues *one* micro-query and may *replace* a distractor at constant $k$.

**Findings & reading.** Increasing $L$ yields a large first-step jump (**+35** pp avg. from $L=0 \rightarrow 1$), then diminishing returns (**+4** pp to $L=3$, **+2** pp to $L=5$); overall **+41** pp at $L=5$ vs. $L=0$. Because $k$ is fixed, gains indicate *targeted replacement* rather than context expansion.

**Implications.** Under tight latency/cost (small $k$), most of SEAL's improvement is captured by $L \in \{1, 3\}$; $L=5$ targets long-tail cases. Full precision/recall/F1@k trends for this ablation are included in the *Supplementary Material*.

## 7 LIMITATIONS & THREATS TO VALIDITY

**LLM-judge variance.** Our primary accuracy metric (*Final Answer Correctness / Judge-EM*) relies on a fixed LLM judge and rubric. Prior work shows LLM judges can exhibit position/verbosity biases and variability (Zheng et al., 2023). We mitigate this by pinning the judge model and decoding, using a deterministic rubric, and reporting paired significance and bootstrap CIs; per-item judgments and prompts are provided in the *supplementary material*. Small effects should be interpreted with caution (Yeh, 2000; Koehn, 2004; Dror et al., 2018).

**Task-coupled answer profile.** For HotpotQA we constrain outputs to short spans with citations (or abstain) to keep judging unambiguous; this favors factoid QA and may under-represent tasks that require long rationales. SEAL is not restricted to short outputs, but transferring to other tasks should pair SEAL with task-appropriate metrics and rubrics.

**Fixed-$k$ by design.** Experiments purposefully use small, fixed retrieval depths ($k \in \{1, 3\}$) to stress *per-slot utility* and isolate loop-driven *replacement* effects. Larger or dynamic $k$ policies are feasible but outside our scope; we leave dynamic-$k$ controllers to future work.

**Domain shift.** All results are on HotpotQA fullwiki (Yang et al., 2018). Performance may differ on other corpora (news, biomedical, legal) or retrieval tasks (fact-checking, argument retrieval). To aid replication beyond Wikipedia and broadened IR settings such as BEIR (Thakur et al., 2021; Petroni et al., 2021), we release predictions, prompts, and scripts in the *supplementary material*.

**Prompt brittleness & component sensitivity.** SEAL's behavior depends on prompts for sufficiency checks, extraction, micro-queries, ranking, and answer emission. Changing judge/backbone or editing prompts can shift outcomes. We therefore pin model IDs/decoding, version prompts/configs, and recommend paired tests/bootstraps when adapting to new domains.

**Seeded slice.** We report results on a seeded 1k HotpotQA validation slice to hold variance constant. Although this facilitates controlled comparisons, it may miss distributional edges. Our released artifacts allow re-running on new 1k samples and regenerating tables with the same evaluation pipeline.

## 8 CONCLUSION

**Conclusion.** We presented **SEAL-RAG** (**S**earch $\rightarrow$ **E**xtract $\rightarrow$ **A**ssess $\rightarrow$ **L**oop), a training-free, inference-time controller for RAG that performs *targeted repairs* under a fixed retrieval depth $k$. SEAL keeps $k$ constant, diagnoses evidence gaps via on-the-fly, entity-anchored extraction, and issues one micro-query per repair until a scope-aware sufficiency check is met. On a seeded 1k *HotpotQA* slice, across four backbones and $k \in \{1, 3\}$, SEAL consistently outperforms **SELF-RAG**: at $k=1$ we observe +10–22 pp Judge-EM gains and sizable Gold-title F1 lifts; at $k=3$ SEAL maintains a +3–13 pp Judge-EM advantage while improving Precision@k by +12–18 pp on average. A loop-budget ablation at $k=1$ shows large first-step benefits (from $L=0$ to $L=1$), with diminishing returns thereafter, indicating that *replacement*—not breadth—drives the gains. Overall, entity-anchored gap repair with a fixed $k$ yields more precise, grounded multi-hop answers without additional training.

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

# A APPENDIX

## A.1 LLM USAGE DISCLOSURE

In accordance with ICLR guidance, we disclose all uses of large language models (LLMs) in preparing this submission. No LLM was used to generate scientific ideas, methods, results, or conclusions. LLM assistance was used *only* for writing-aid and polishing tasks, such as: (i) grammar and style edits on author-written text; (ii) rephrasing sentences for clarity and concision; and (iii) converting author-provided bullet points into prose *without adding technical content*. All technical sections (problem setup, method, experiments, metrics, analyses, and conclusions) were written and verified by the authors. All tables, figures, numbers, and claims are based on code and logs produced by our implementation and were independently checked by the authors.

## A.2 SUPPLEMENTARY MATERIAL

We provide an anonymized `.zip` archive containing all artifacts required to reproduce the results. **Contents:**

- **Code:** scripts to run SEAL-RAG and **SELF-RAG** baselines on the seeded HotpotQA slice; evaluation scripts to recompute all tables/figures; prompt templates for retrieval, micro-queries, sufficiency checks, ranking, judging, and answer emission.

- **Prediction files:** per-example outputs for every backbone $\times$ $k \in \{1, 3\}$ $\times$ method (JSONL/CSV), enabling exact regeneration of reported metrics.

- **Metrics notebooks:** one-click notebooks to recompute Judge-EM and Gold-title Precision/Recall/F1@k, plus plots.

- **Statistical tests:** scripts for $\chi^2$ on paired Judge-EM outcomes and paired two-sided $t$-tests for Precision/Recall/F1, with Holm–Bonferroni correction; bootstrap CI utilities.

- **Index manifest:** Wikipedia dump date, doc IDs/titles, embedding model/dimension, distance metric, and index build parameters; deterministic seed and chunking settings.

- **Seeded slice:** the list of 1,000 HotpotQA validation QIDs and the RNG seed used for sampling.

