# OpenReview forum: "SEAL-RAG: Loop-Adaptive RAG with On-the-Fly Entity Extraction and Fixed-k Gap Repair"
_ICLR.cc/2026/Conference — ICLR 2026 Conference Withdrawn Submission_

### Official Review · Reviewer_nUdh · 2025-10-16

**Soundness:** 2
**Presentation:** 2
**Contribution:** 2
**Rating:** 4
**Confidence:** 4

**Summary:**

This paper introduces SEAL-RAG, a training-free, inference-time controller for RAG. It operates on a Search → Extract → Assess → Loop cycle with a fixed retrieval context size. The core idea is to use on-the-fly entity extraction to identify specific information gaps and then issue targeted micro-queries to fill them. Unlike methods that expand the context, SEAL-RAG replaces less relevant documents with newly retrieved ones, maintaining a fixed cost profile.

**Strengths:**

The paper tackles the important and practical problem of improving RAG performance under a constrained budget. Besides, the method demonstrates significant and consistent performance gains over SELF-RAG on the chosen task in terms of answer correctness and evidence precision.

**Weaknesses:**

1. The core contribution appears to be a procedural and technical refinement of the iterative, self-correcting RAG paradigm pioneered by models like SELF-RAG. The overall framework about retrieving, assessing, and re-retrieving is not new. The work feels more like an engineering improvement focused on explicit entity-based control, rather than a fundamental conceptual advance.
2. The evaluation is confined to a single dataset, HotpotQA. This raises questions about the method's generalizability.
3. The paper introduces several new modules (e.g., entity-anchored extraction, scope-aware sufficiency gate) as improvements over SELF-RAG's mechanisms. However, the experiments do not offer an ablation or comparative analysis to demonstrate the specific effectiveness of these modules.
4. The paper introduces a complex control loop with many new, specific terms (e.g., "gap specification," "scope-aware sufficiency", "entity-first ranking"). This density can make the methodology difficult to follow and obscures the core mechanics.

**Questions:**

Why use 1,000-example HotpotQA rather than the complete HotpotQA?

---

> ### Author Response · Authors · 2025-12-03
> **Author comment on Reviewer nUdh’s review**
>
> Thank you very much for the thoughtful and balanced review, and for recognizing both the practical importance (fixed-budget RAG) and the consistent gains over SELF-RAG.
>
> Conceptual vs. procedural contribution:
> We agree that the high-level paradigm of “retrieve → assess → re-retrieve” is not new. Our intended contribution is how this is instantiated: SEAL-RAG explicitly treats multi-hop RAG as entity-anchored gap repair under a fixed evidence budget. Concretely, we (i) maintain an entity ledger extracted from the current context, (ii) formulate a typed gap specification over missing bridge entities/relations, corroboration/contradiction, and answerability, and (iii) issue micro-queries that replace distractors within a fixed-k context. We will make this framing much more explicit in the introduction and method overview, so the core idea does not get buried under terminology.
>
> Single dataset + 1,000-example subset:
> You are absolutely right that a single dataset limits generality. For this submission we focused on HotPotQA and a 1,000-example subset to make it feasible to run multiple backbones and loop budgets under API-based constraints and to analyse behaviour in detail. In follow-up work we are already scaling to the full HotPotQA dev set and adding MuSiQue and 2WikiMultiHopQA, so that generalization beyond this subset and dataset can be properly assessed; we will clarify this motivation and plan in the text.
>
> Module ablations and complexity:
> We agree that the current experiments under-analyze the individual modules. In a revised version, we plan to add ablations that (i) remove the entity ledger / gap specification (falling back to doc-level critics), (ii) replace the scope-aware sufficiency gate with a simpler scalar decision, and (iii) switch from replace-don’t-expand to an expand-only variant. We also take your presentation concern seriously: we will streamline the terminology, collapse overlapping terms, and add a small schematic of the control loop so the mechanics are easier to follow.
>
> We appreciate your constructive feedback; it has helped us better separate the core idea (fixed-budget, entity-anchored gap repair) from the surrounding engineering details.

---

### Official Review · Reviewer_byQa · 2025-10-24

**Soundness:** 2
**Presentation:** 1
**Contribution:** 2
**Rating:** 2
**Confidence:** 5

**Summary:**

This paper proposes a plug-and-play style RAG method, SEAL-RAG, which performs a Search, Extract, Assess, and then Loop cycle. The experiments show the proposed method outperforms a specific existing method on a QA dataset. Their ablation study shows an iterative repair loop improves the performance.

**Strengths:**

- This paper proposes SEAL-RAG that uses on-the-fly entity extraction to build an explicit gap specification. This allows it to issue focused micro-queries to find specific missing facts for next retrieval.
- The proposed method keeps the retrieval depth fixed and replaces distractors with newly retrieved passages that seem more important.
- Since SEAL-RAG is a training-free, inference-time controller, it is easy to be combined with any LLMs.

**Weaknesses:**

- The experiments rely solely on HotpotQA. To demonstrate robustness and generality for multi-hop QA, it would be better to include additional benchmarks such as MuSiQue [1] and 2WikiMultiHopQA[2].
- Several core experiments fix k=1, i.e., retrieving a single chunk, despite the paper’s focus on multi-hop questions. Given the nature of multi-hop reasoning, evaluating larger k, e.g., k \in [5, 7, 10], would be more natural and enable fairer comparisons with prior work.
- If I understand correctly, Self-RAG [3] provides released fine-tuned 7B and 13B variants. The manuscript, however, states that the backbone is interchangeable and not fine-tuned, and I checked the supplementary material and found the authors used the implementation from LangChain, which uses OpenAI API, i.e., not fine-tuned models. No explanation about this is provided in the paper.
- Since the paper’s scope excludes fine-tuning of retriever, reranker, or generator, plug-and-play approaches seem better matched than Self-RAG for primary comparison. Consider adding Adaptive-RAG [4], which shows better performance than Self-RAG, Adaptive-k [5], and LC-Boost [6]. All of these include HotpotQA, enabling direct comparisons.

[1] Trivedi, Harsh, et al. "♫ MuSiQue: Multihop Questions via Single-hop Question Composition." Transactions of the Association for Computational Linguistics 10 (2022): 539-554.

[2] Ho, Xanh, et al. "Constructing A Multi-hop QA Dataset for Comprehensive Evaluation of Reasoning Steps." *Proceedings of the 28th International Conference on Computational Linguistics*. 2020.

[3] Asai, Akari, et al. "Self-RAG: Learning to Retrieve, Generate, and Critique through Self-Reflection." The Twelfth International Conference on Learning Representations.

[4] Jeong, Soyeong, et al. "Adaptive-RAG: Learning to Adapt Retrieval-Augmented Large Language Models through Question Complexity." Proceedings of the 2024 Conference of the North American Chapter of the Association for Computational Linguistics: Human Language Technologies (Volume 1: Long Papers). 2024.

[5] Taguchi, Chihiro, Seiji Maekawa, and Nikita Bhutani. "Efficient Context Selection for Long-Context QA: No Tuning, No Iteration, Just Adaptive-$ k$." arXiv preprint arXiv:2506.08479 (2025).

[6] Qian, Hongjin, et al. "Are Long-LLMs A Necessity For Long-Context Tasks?." arXiv preprint arXiv:2405.15318 (2024).

**Questions:**

Could you answer all the bullet points I raised in the Weaknesses section?

---

> ### Author Response · Authors · 2025-12-03
> **Author comment on Reviewer byQa’s review**
>
> Thank you very much for your careful and constructive review, and for highlighting both the strengths (entity-anchored gap specification, fixed-depth replacement, training-free controller) and the missing pieces. I address each weakness point-by-point below.
>
> 1. Only HotPotQA (MuSiQue, 2WikiMultiHopQA):
> We agree that relying solely on HotPotQA is limiting. For this initial submission we chose HotPotQA as a canonical, well-understood bridge-style benchmark and focused on analysing SEAL-RAG’s behaviour there under fixed evidence budgets. In ongoing work we are already extending experiments to MuSiQue and 2WikiMultiHopQA, and in a revised version we will (i) add results on these datasets and (ii) discuss where the entity-anchored gap specification helps or struggles relative to their more diverse reasoning structures.
>
> 2. Core experiments at k = 1:
> You are correct that several core experiments fix k=1. This was intentional to stress-test the tightest evidence-budget regime, where replacement of distractors is most critical, but we agree that multi-hop QA normally uses larger k and that this choice should be justified and complemented with larger-k results. We already include comparisons at k=3; in a revised version we plan to add experiments for k∈{5,7,10} and show how SEAL-RAG’s advantage decays as k increases, making the “low-k focus” explicit rather than implicit.
>
> 3. Self-RAG implementation (fine-tuned vs LangChain):
> You are absolutely right that the Self-RAG paper releases fine-tuned 7B/13B models. In our experiments, however, we intentionally used the LangChain / LangGraph-style Self-RAG flow with OpenAI models, i.e., a training-free self-reflective controller implemented via prompting, to keep all methods in the same “no fine-tuning” regime. We acknowledge that this was not clearly explained; we will explicitly rename this baseline (e.g., “Self-reflective RAG, LangGraph implementation”) and clarify that no Self-RAG fine-tuning is used.
>
> 4. Additional plug-and-play baselines (Adaptive-RAG, Adaptive-k, LC-Boost):
> We fully agree that these are highly relevant plug-and-play comparators and that the paper should situate SEAL-RAG more clearly with respect to them. Conceptually, Adaptive-RAG learns a classifier over question complexity and routes each query to an appropriate retrieval strategy (no retrieval, single-step RAG, multi-step RAG), optimizing which pipeline to use. Adaptive-k is a single-pass, training-free method that dynamically chooses the number of retrieved passages based on the similarity-score distribution, optimizing how many passages to keep without iterative loops. LC-Boost targets long-context tasks by using a short-context model that iteratively decides which parts of a long input to read next, rather than operating in an external-corpus multi-hop QA setting. In contrast, SEAL-RAG assumes a fixed retrieval depth and treats multi-hop RAG as entity-anchored gap repair under a strict evidence budget, using an explicit entity ledger and replace-don’t-expand micro-queries. In follow-up work we plan to (i) add these methods to the related work section with this distinction made explicit, and (ii) include direct empirical comparisons where feasible using their public implementations and reported HotPotQA results.
>
> We appreciate your detailed pointers and will revise the paper to make SEAL-RAG’s evaluation scope, baseline choices, and positioning within this literature much clearer.

---

### Official Review · Reviewer_BwqJ · 2025-11-02

**Soundness:** 2
**Presentation:** 2
**Contribution:** 1
**Rating:** 2
**Confidence:** 4

**Summary:**

This paper introduces SEAL-RAG, a clever, training-free controller designed to make RAG better at handling complex, multi-hop questions. Instead of just retrieving a bunch of documents and hoping for the best, SEAL-RAG uses a smart Search → Extract → Assess → Loop cycle. What's really neat is that it operates on a fixed budget of retrieved documents (k), identifying specific missing facts by extracting entities on the fly and then issuing targeted "micro-queries" to fill those gaps. This process replaces less useful documents with more relevant ones rather than just expanding the context. The authors show on a HotpotQA benchmark that this approach significantly improves answer correctness and evidence precision over a strong baseline like SELF-RAG, especially when the retrieval budget is tight.

**Strengths:**

1. Its "replace, don't expand" strategy is a very intuitive way to fight context dilution by actively swapping out distractors for crucial bridge facts.

2. Being a training-free controller that works with existing models makes the method highly practical and easy for others to adopt.

**Weaknesses:**

1. The whole approach feels very tailored to the entity-centric, factoid style of HotpotQA and might not generalize well to tasks requiring more abstract reasoning.

2. The system's performance seems to hinge on the entity extraction and sufficiency check prompts working perfectly, which could be quite brittle in practice.

3. The performance gains, while impressive, seem to shrink as the base models get stronger and the retrieval budget (k) increases.

**Questions:**

It looks like the sufficiency checks and micro-query generation rely heavily on well-crafted prompts. How sensitive is SEAL-RAG to the specific wording of these prompts, and was there a lot of manual prompt engineering involved to achieve these results?

---

> ### Author Response · Authors · 2025-12-03
> **Author comment on Reviewer BwqJ’s review**
>
> Thank you very much for the thoughtful and encouraging review, and for clearly articulating both the strengths and the concerns.
>
> 1. Scope and generalization:
> You are right that SEAL-RAG is currently evaluated only on HotPotQA and that the design leans into its entity-centric, factoid structure. Our intention, however, is broader: we view SEAL-RAG as a general training-free controller for fixed-budget, gap-repair retrieval, and HotPotQA is our first testbed. In ongoing work we are extending experiments to MuSiQue and 2WikiMultiHopQA, which feature more diverse reasoning patterns, and we will clarify in the paper that HotPotQA is a starting point rather than the only intended domain. We also agree that more abstract, non-entity-centric tasks are an important stress test, and plan to include at least one such dataset in a revised version.
>
> 2. On k, model strength, and where SEAL-RAG helps:
> We agree that the largest gains appear in the low-k and mid-tier model regimes. This is somewhat by design: SEAL-RAG is aimed at tight evidence budgets, where replacing distractors with bridge facts matters most. On HotPotQA in particular, each question is constructed from a small number of relevant Wikipedia pages, so as k grows, even standard baselines are increasingly likely to retrieve both gold pages and close much of the gap. As base models and k increase, the baseline’s own reasoning and this “more shots at the gold pages” effect naturally reduce the available headroom. We will make this positioning explicit and add curves that show how the benefit decays as k and model size grow.
>
> 3. Prompt sensitivity and brittleness:
> The sufficiency checks and micro-queries indeed rely on prompting. In this submission we used simple, shared templates across all models and did not perform heavy per-model prompt tuning. We did not conduct a full sensitivity sweep, and we agree that a more systematic analysis is needed; in future work we plan to (i) report performance under several alternative sufficiency / extraction templates, and (ii) quantify how much of the gain comes from the controller logic versus prompt wording. We will also emphasize that the same prompts are reused across datasets and backbones, to reduce the impression of brittle, hand-crafted engineering.
>
> We appreciate your positive assessment of the “replace, don’t expand” strategy and the practicality of a training-free controller, and we will revise the paper to better reflect these intended design goals and limitations.

---

### Official Review · Reviewer_sPfk · 2025-11-04

**Soundness:** 2
**Presentation:** 2
**Contribution:** 1
**Rating:** 2
**Confidence:** 4

**Summary:**

The authors propose a pipeline for multi-hop retrieval-augmented generation. Without training any components, it iterates retrieval, entity extraction, and a ranking of evidence. The method is compared under retrieving k=1 or k=3 passages per hop against some implementation of Self-RAG (unclear if that implementation uses training, like the original Self-RAG) and finds higher accuracy and retrieval quality in both cases.

**Strengths:**

The method proposed performs much better on HotPotQA than the authors' baseline. This is also tested for statistical significance.

**Weaknesses:**

It is difficult to identify a specific novelty in the proposed methods or to ascertain if it is empirically superior to prior work.

The method combines a number of very standard techniques and does not offer a compelling reason for several of them. The techniques range from fairly natural (e.g., the repair check, which is fundamentally just a self-stopping criteria, an extremely standard component in these types of pipelines) to fairly surprising but not fundamentally justified in a new informative way (e.g., explicit entity extraction with modern LLMs).

The comparison against some implementation of Self-RAG raises more questions than it answers. Why Self-RAG in particular? While it is a reasonable system it is neither a particularly state-of-the-art or particularly popular/fundamental baseline, and --- if I understand correctly --- it fundamentally centers on training the models, whereas my understanding is that the authors test both methods without any training.

The authors evaluate only on HotPotQA. While HotPotQA is a fairly foundational dataset in this space, there have been numerous (harder) benchmarks in the space of multi-hop retrieval and "deep research" over the past 7 or so years since the introduction of this task. HotPotQA has over 3600 citations. Many of the methods evaluated in the literature on HotPotQA are also training-free and achieve a range of higher scores. In what way does the proposed method differ from or compare with the vast literature on this topic?

**Questions:**

See weaknesses.

---

> ### Author Response · Authors · 2025-12-03
> **Clarification on novelty, baselines, and scope (to Reviewer sPfK)**
>
> 1. Novelty and design choices:
> Our goal is not to claim novelty for each individual module, but for the controller and problem framing. SEAL-RAG treats multi-hop RAG under deployment constraints as a fixed-evidence gap-repair problem: the system must answer using at most k passages, and additional computation is used only to repair specific gaps in that fixed budget. To do so, SEAL-RAG (i) maintains an entity-anchored ledger of verbatim entities/relations, (ii) defines a gap specification over bridging entities, corroboration/contradiction and answerability, and (iii) performs gap-conditioned micro-queries that replace distractors within the same k slots rather than expanding context. We agree that self-stopping and entity extraction are known tools, but the way they are combined into a fixed-k, replacement-only controller is what we see as the main contribution, and we will make this framing much clearer.
>
> 2. Why Self-RAG and what implementation:
> We apologize for the confusion: we do not use the original fine-tuned Self-RAG models. Instead, we follow the training-free LangGraph-style Self-RAG flow, where a frozen LLM grades document relevance, hallucinations, and answer usefulness and may rewrite the query. We chose this to ensure training parity: all compared methods are training-free controllers over the same backbone, rather than mixing fine-tuned and non-fine-tuned models. In the camera-ready, we will rename this baseline (“Self-reflective RAG, LangGraph implementation”) and clarify this point.
>
> 3.Evaluation scope and prior work:
> We agree that evaluating only on HotPotQA and only against our own baseline under-represents the broader literature. In follow-up work we plan to (i) add MuSiQue and 2WikiMultiHopQA, and (ii) include stronger training-free baselines (e.g., Adaptive-k / set-selection and CRAG-style corrective RAG) and module-level ablations, so that SEAL-RAG is more clearly positioned relative to this body of work.

---

### Note · Authors · 2025-12-11

I have read and agree with the venue's withdrawal policy on behalf of myself and my co-authors.